# Sustainable Rehabilitation of Post-Bauxite Mining Land for *Albizia falcata* Cultivation Using Specific Location Amelioration Technology

Denah Suswati *[ID] and Nugra Irianta Denashurya

Faculty of Agriculture, Tanjungpura University, Pontianak 78124, Indonesia
* Correspondence: denah.suswati@faperta.untan.ac.id

**Abstract:** Bauxite mining, catalyzed by the escalating global demand for aluminum, leads to serious environmental repercussions, necessitating the development of efficient land rehabilitation techniques. This study presents a sustainable solution for post-bauxite mining land rehabilitation by leveraging red mud waste and cow manure fertilizer. Applied in PT Antam, Sanggau Regency, West Kalimantan, this research examines the potential of these ameliorants to restore ecological functions and promote the growth of *Albizia falcata* plants. Our findings reveal a remarkable enhancement in soil pH levels and nutrient availability (N, P, K, Ca, Mg, and Na) when applying a blend of 10% red mud and 20% cow manure fertilizer. Consequently, a significant improvement in the growth of *Albizia falcata* plants by factors ranging from 8 to 13 times was recorded. These results, alongside potential economic benefits, highlight the promise of this approach not to only confront the challenges posed by bauxite mining but also to contribute to global land rehabilitation strategies. While this study provides substantial insights, it recommends further exploration of this method involving diverse plant species, treatments with different ameliorants, and a broader range of observed variables. The study underscores the critical role of government intervention through stringent regulations and the need for a more comprehensive environmental and cost-benefit analysis to foster sustainable mining practices and responsible land rehabilitation.

**Keywords:** bauxite mining; post-mining land rehabilitation; red mud waste and cow manure fertilizer; *Albizia falcata* cultivation; sustainable mining practices

## 1. Introduction

The expanding global demand for aluminum has catalyzed a surge in bauxite mining across the globe [1,2]. This mining process necessitates the removal of vegetation, topsoil, and overburden layers, yielding severe environmental repercussions, including soil degradation, habitat alterations, and landscape transformations [3,4]. Issues stemming from bauxite mining frequently include ecosystem damage, water and air pollution, landscape degradation, waste management difficulties, and socio-economic impacts [5,6].

Recent academic research has underscored the importance of comprehensive sustainable mining strategies in confronting environmental issues linked with mining operations, notably in deep mines. Dong et al. (2019) stressed the role of cleaner production, an integrative preventive environmental approach, in reducing the ecological impacts of mining [7]. This research brought to light developments and new perspectives on environmental issues and deep mining strategies for cleaner production in mines. Their investigations offered a summary of the general impacts of the mining industry on the ecological environment, presented remedies for ecological contamination caused by tailings dams, and recommended distinct strategies for safer and more effective resource extraction in deep mines. In a parallel vein, Cai et al. (2021) introduced vital engineering technologies intended for green, intelligent, and sustainable development in deep metal mines [8]. This research drew attention to the engineering hurdles that arise in the deep mining of metal mineral

resources and suggested a variety of technologies and innovative methods to address these challenges.

Despite these challenges, the economic significance of bauxite mining, including providing essential mineral resources such as aluminum, creating employment opportunities, and fostering innovation and technological advancement, cannot be understated [9–13]. Moreover, efficient post-mining land rehabilitation is imperative for ensuring the sustainability of mining activities [14,15]. A solitary bauxite mine can impact over 100 hectares of land annually, thereby underlining the urgent need for stringent regulations and intervention strategies to mitigate the substantial environmental effects [16,17].

Post-mining land rehabilitation techniques are instrumental in restoring ecological functions, preventing additional environmental harm, adhering to regulations, and promoting sustainable environmental conservation [18–20]. Such techniques can facilitate the restoration of former mining sites to their natural state, recovering soil productivity, reinstating biodiversity, and balancing water and nutrient cycles, thus aligning with the principles of sustainable development and responsible mining practices [21,22].

To tackle the environmental hurdles associated with post-mining land, this research introduces a rehabilitation method that leverages red mud waste and organic materials for land improvement [23]. Red mud waste, a byproduct of bauxite ore refinement, when amalgamated with organic matter like cow manure fertilizer, can efficaciously augment soil pH and enhance nutrient retention [24–26]. This method also harbors economic benefits such as improved availability of wood raw materials, decreased production costs, novel business opportunities, enhanced business sustainability and reputation, and diversified business portfolios, thus addressing the scarcity of wood resources and curtailing environmental damage incurred through excessive limestone exploitation [27–29].

The present study not only devises pragmatic solutions to confront economic, social, and environmental challenges related to bauxite mining but also contributes to the development of efficient land rehabilitation techniques applicable globally. By adhering to regulations and implementing these techniques, mining firms can effectively and efficiently meet their land rehabilitation obligations, bringing benefits to both the environment and local communities [30,31]. Therefore, the appropriate management of post-mining land is integral to mitigating adverse environmental impacts and fostering sustainable mining practices, underscoring the crucial role of governmental intervention via stringent regulations given the considerable macro-environmental implications.

## 2. Materials and Methods

This year-long study was undertaken on the premises of PT Antam in Sanggau Regency, West Kalimantan, with soil chemical analyses executed by the Soil Chemistry and Fertility Laboratory of the Faculty of Agriculture at Tanjungpura University. The research implemented a factorial arrangement in a completely randomized block design, incorporating eight treatments. These treatments included R0 (100% Ultisol soil), R1 (95% Ultisol soil + 5% red mud), R2 (90% Ultisol soil + 5% red mud + 5% cow manure), R3 (85% Ultisol soil + 5% red mud + 10% cow manure), R4 (80% Ultisol soil + 10% red mud + 15% cow manure), R5 (70% Ultisol soil + 10% red mud + 20% cow manure), R6 (70% Ultisol soil + 15% red mud + 15% cow manure), and R7 (60% Ultisol soil + 15% red mud + 25% cow manure).

In this experiment, a total of 128 *Albizia falcata* seedlings were engaged. Eight treatments were devised, each comprising four replications, with each replication including six seedlings. These seedlings were systematically planted at an interval of 3 × 3 m, leading to the presence of 24 *Albizia falcata* seedlings within each treatment group. Subsequently, the *Albizia falcata* seedlings were cultivated on post-bauxite mining land in alignment with the predetermined planting medium treatments. To promote the growth of these seedlings, organic pots were utilized as containers for the planting medium. The composition of these organic pots consisted of a blend of 80% cow manure compost and 20% red mud. Thereafter, planting medium mixtures corresponding to each treatment (eight in total) were filled into

the organic pots. The planting medium within these organic pots was then incubated for a span of two weeks to adequately prepare the medium prior to the planting of the *Albizia falcata* seedlings.

Sequentially, soil samples were extracted for chemical properties analysis. Soil pH was ascertained via a pH meter with pH 7.0 and pH 4.0 buffer solutions [32], while the organic carbon content was established via the Walkey and Black wet oxidation method [33]. Total nitrogen content was quantified employing the Kjeldahl method [34], available phosphorus through the Bray-I method [35], and exchangeable potassium ($K^+$), calcium ($Ca^{2+}$), magnesium ($Mg^{2+}$), and sodium ($Na^+$) using the NH4Oac 1 N pH 7 extraction method [36]. The cation exchange capacity (CEC) was evaluated using the indophenol blue method [37].

Subsequently, *Albizia falcata* seedlings, two months old and approximately 62 cm in height, were planted in each organic pot. Growth parameters, encompassing plant height and stem diameter, were measured upon the plants' attaining 10 weeks of age. The gathered data were then subjected to statistical analysis, utilizing the F-test and Duncan's multiple range test (DMRT) at a 5% significance level [38].

## 3. Results and Discussion

### 3.1. Description of Soil Fertility in Post-Mining Bauxite Tailings

The post-mining bauxite tailings soil is predominantly composed of gravel with a small amount of sand, and the upper layer is mostly gravel, making it challenging for vegetation to thrive [28,39]. Vegetation that does manage to grow in such areas tends to be sparse, dry, and stunted. To enhance the physical properties of the soil in former bauxite tailings, it is recommended to plant fast-growing species with a high leaf count. The decomposition of stems, branches, and dead leaves, which integrate with the soil, can improve soil fertility and enhance its physical properties [40,41].

An effective method to increase soil pH and nutrient availability is through the addition of red mud. Red mud not only provides these benefits but also enhances several physical properties of the soil due to its clay content of 36.58%. Table 1 presents some of the chemical properties of red mud that contribute to these improvements.

**Table 1.** Chemical properties of the soil, red mud, and cow manure before treatment.

| Chemical Parameters | Post-Mining Bauxite Tailings Soil | Red Mud | Cow Manure Compost |
|---|---|---|---|
| Texture | | | |
| Sand (%) | 24.20 (%) | 10.87 (%) | - |
| Silt (%) | 32.57 (%) | 52.55 (%) | - |
| Clay (%) | 43.23 (%) | 36.58 (%) | - |
| pH | 3.34 | 10.14 | 6.60 |
| Organic-C | 0.62 (%) | 0.18 (%) | 34.22 (%) |
| N-total | 0.07 (%) | 0.03 (%) | |
| Available P | 4.83 (ppm) | 0.52 (ppm) | 0.17 (%) |
| Exch. K | 0.03 (cmol(+)kg$^{-1}$) | 0.13 (cmol(+)kg$^{-1}$) | 0.27 (%) |
| Exch. Ca | 0.94 (cmol(+)kg$^{-1}$) | 4.22 (cmol(+)kg$^{-1}$) | 0.59 (%) |
| Exch. Mg | 0.52 (cmol(+)kg$^{-1}$) | 0.18 (cmol(+)kg$^{-1}$) | 0.25 (%) |
| Exch. Na | 0.03 (cmol(+)kg$^{-1}$) | 0.67 (cmol(+)kg$^{-1}$) | - |
| CEC | 42.33 (cmol(+)kg$^{-1}$) | 5.37 (cmol(+)kg$^{-1}$) | - |
| Base Saturated | 3.59 (%) | 96.83 (%) | - |

Source: analysis results.

Table 1 presents the pH values of red mud from PT. ICA, which can be used in various types of soil. Red mud not only serves as a nutrient source but also helps maintain a balanced nutrient profile in the soil. Its application is expected to enhance soil pH and base saturation in post-bauxite mining land, promoting the growth of *Albizia falcata*.

### 3.2. Soil Nutrient Availability

Table 2 provides the results of soil nutrient availability analysis for N, P, K, Ca, Mg, and Na due to the influence of red mud and cow manure treatments.

**Table 2.** Soil pH and nutrient availability of N, P, K, Ca, Mg, and Na after incubation with red mud and cow manure treatments.

| Treatment | Soil Analysis Results | | | | | | | | | |
|---|---|---|---|---|---|---|---|---|---|---|
| | pH | Organic-C | N-Total | Available P | Exch. K | Exch. Ca | Exch. Mg | Exch. Na | CEC | BS |
| R0 | 4.26 g | 0.22 f | 0.03 f | 13.49 g | 1.10 d | 0.6 e | 0.24 e | 2.25 e | 8.98 e | 46.66 e |
| R1 | 7.53 d | 2.43 e | 1.28 d | 22.83 f | 1.17 d | 4.81 d | 2.65 d | 5.21 d | 10.43 d | 132.69 d |
| R2 | 8.03 c | 2.77 d | 1.27 d | 32.39 e | 1.59 c | 12.45 c | 3.20 c | 5.86 c | 12.26 c | 188.42 c |
| R3 | 8.26 b | 2.99 c | 1.37 c | 39.03 d | 1.71 bc | 17.82 b | 3.43 bc | 7.91 b | 12.51 bc | 246.76 b |
| R4 | 8.35 b | 2.98 c | 1.51 b | 41.32 c | 2.07 ab | 19.98 b | 3.50 bc | 8.15 ab | 12.79 ab | 263.49 b |
| R5 | 8.42 a | 3.69 a | 1.82 a | 44.28 b | 2.38 a | 25.55 a | 3.64 bc | 8.22 ab | 13.54 a | 293.87 a |
| R6 | 7.25 e | 3.17 b | 1.28 d | 43.93 b | 1.81 bc | 19.25 b | 3.98 a | 8.30 ab | 13.03 a | 255.87 b |
| R7 | 7.13 f | 3.21 b | 0.76 e | 46.29 a | 1.61 bc | 17.21 b | 3.91 a | 8.01 ab | 13.07 a | 235.20 b |

Remarks: The mean followed by the same letter indicates no significant difference in the same row. Source: analysis results.

### 3.2.1. Reaction (pH) in Soil after Incubation

Table 2 demonstrates that treatment R5 (combination of 70% Ultisol soil + 10% red mud + 20% cow manure) significantly influenced soil pH after incubation compared to other treatments. However, treatment R7 (a combination of 60% Ultisol soil + 15% red mud + 25% cow manure) resulted in a decrease in soil pH. This decrease can be attributed to the higher amount of red mud and cow manure added in treatment R7. The decomposition of cow manure releases organic acids, contributing $H^+$ ions to the soil and leading to a decrease in pH [42,43].

Increasing the amounts of red mud and cow manure can raise soil pH after incubation. Red mud has a pH $H_2O$ of 10.14, and cow manure has a pH $H_2O$ of 6.6. Therefore, increasing the amount of red mud to 10% and cow manure to 20% added to the soil can significantly increase the pH of Ultisol soil. Moreover, red mud contains base cations, which can increase the percentage of base saturated (BS) in the colloidal complex, directly causing an increase in soil pH. Figure 1 illustrates the effect of the combination of red mud and cow manure on soil pH.

### 3.2.2. Organic Carbon in Soil after Incubation

Table 2 indicates that increasing the amount of cow manure added up to treatment R7 (combination of 60% Ultisol soil + 15% red mud + 25% cow manure) can enhance soil organic carbon (C-organic) compared to the control without the addition of red mud and cow manure. Although red mud has low levels of C-organic, when combined with cow manure, which has a high C-organic content of 34.22%, it can increase soil organic carbon. Figure 2 presents the effect of the combination of red mud and cow manure on soil organic carbon.

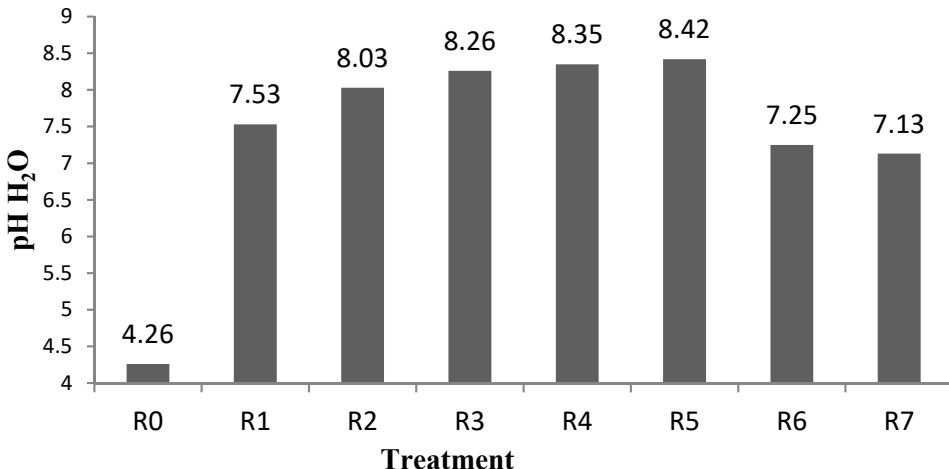

**Figure 1.** Soil pH after incubation with red mud and cow manure treatments.

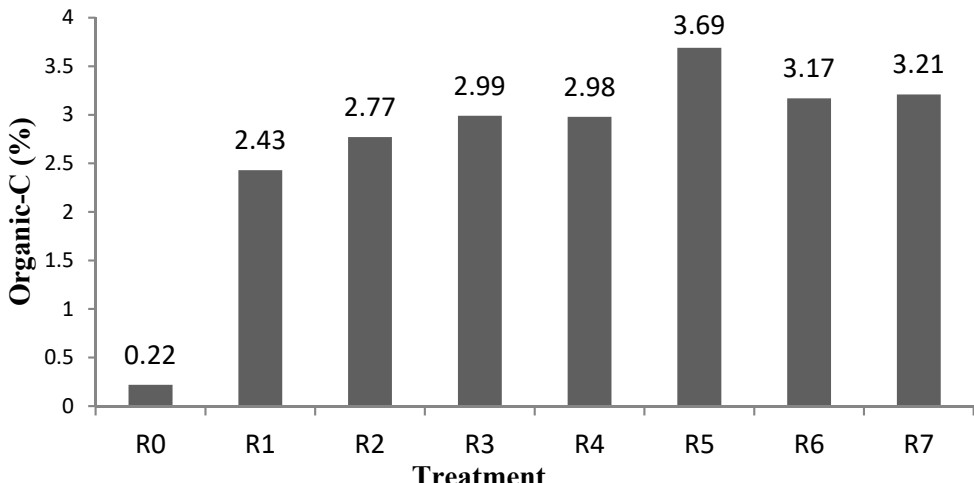

**Figure 2.** Soil organic carbon after incubation with red mud and cow manure treatments.

### 3.2.3. Total Nitrogen in Soil after Incubation

Table 2 reveals that the addition of red mud and cow manure can increase total soil nitrogen compared to the control soil, which had a nitrogen content of 0.03%. Treatment R5 (combination of 70% Ultisol soil + 10% red mud + 20% cow manure) resulted in the highest increase in total soil nitrogen after incubation compared to other treatments. However, increasing the amount of red mud and cow manure in treatment R6 (70% Ultisol soil + 15% red mud + 15% cow manure) and treatment R7 (60% Ultisol soil + 15% red mud + 25% cow manure) led to a decrease in total soil nitrogen.

The decrease in total soil nitrogen due to increasing amounts of red mud and cow manure is likely caused by a decrease in soil organic carbon, which slows down the mineralization processes. Nitrogen (N) is mainly derived from the decomposition of organic matter, and its contribution depends on the quality and quantity of the organic matter [44]. Decomposition rates are affected by soil pH, and an increase in pH can decrease nitrogen availability [45–47]. The contribution of nitrogen from air fixation in the soil is relatively low, around 117 kg/ha/year [48]. Organic fertilizers undergo hydrolysis, mineralization, and nitrification processes, reaching maximum accumulation of N-NO3- after 14 days of incubation [49–51]. Figure 3 depicts the effect of the combination of red mud and cow manure on total soil nitrogen.

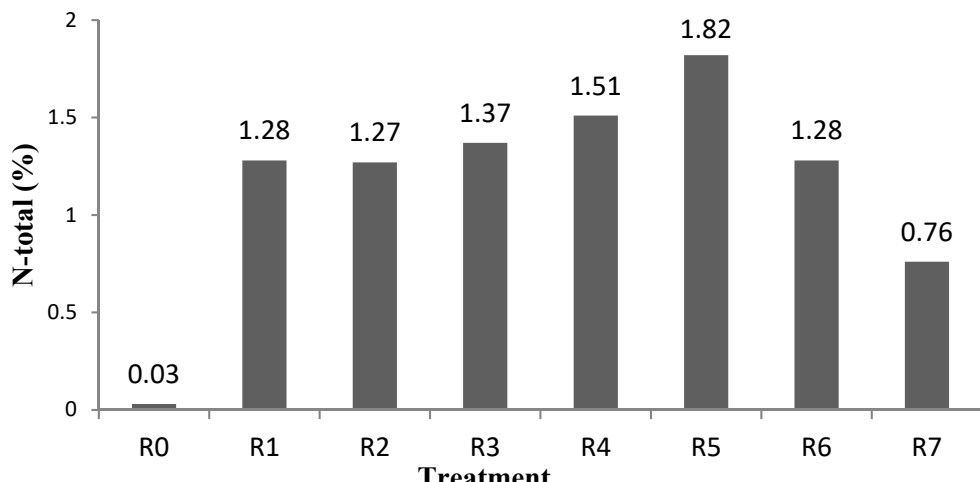

**Figure 3.** Total soil nitrogen after incubation with the combination of red mud and cow manure treatments.

### 3.2.4. Phosphorus in Soil after Incubation

Table 2 demonstrates that the addition of red mud and cow manure can increase available phosphorus in the soil compared to the control soil, which had an available phosphorus content of 13.49 ppm. The addition of increasing amounts of red mud and cow manure in treatment R7 (60% Ultisol soil + 15% red mud + 25% cow manure) resulted in the highest increase in available phosphorus (46.29 ppm) compared to other treatments.

The decrease in available phosphorus in other treatments may be due to an increase in soil pH, which enhances the solubility of base cations, particularly calcium (Ca) ions. This can result in the binding of phosphate ions to calcium ions, leading to reduced phosphorus availability [52,53]. Additionally, high levels of organic matter from cow manure can promote phosphorus fixation and reduce its availability [54]. Figure 4 illustrates the effect of the combination of red mud and cow manure on available phosphorus.

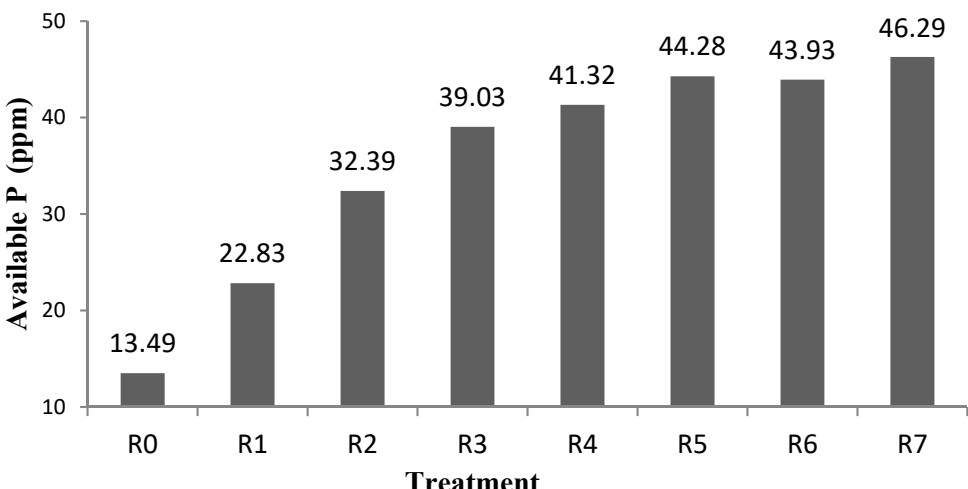

**Figure 4.** Displays the available P in the soil after incubation with the combination of red mud and cow manure treatments.

### 3.2.5. Potassium in Soil after Incubation

Table 2 shows that the application of red mud and cow manure fertilizer can increase the exch. K in the soil compared to the untreated soil. The exch. K content in the untreated soil was 1.10 cmol kg (+) kg$^{-1}$, while the exch. K content after treatment ranged from 1.17 cmol kg (+) kg$^{-1}$ to 2.34 cmol kg (+) kg$^{-1}$. The application of red mud and cow manure fertilizer gradually increased up to the R5 treatment (combination of 70% Ultisol

soil + 10% red mud + 20% cow manure fertilizer), which showed the highest exch. K in the soil. However, it was not significantly different from the R4 treatment (80% Ultisol soil + 10% red mud + 10% cow manure fertilizer). This is because red mud and cow manure fertilizer contain K, which increases the exch. K in the soil upon addition [55,56]. Moreover, the high proportion of clay minerals in the added red mud (36.58%) provides a significant potential for the exch. K in the soil.

In addition to being a source of K nutrients, the high proportion of clay minerals in red mud added to the soil provides great potential for exch. K. The effect of the combination of red mud and cow manure fertilizer treatments on the exch. K in the soil is shown in Figure 5.

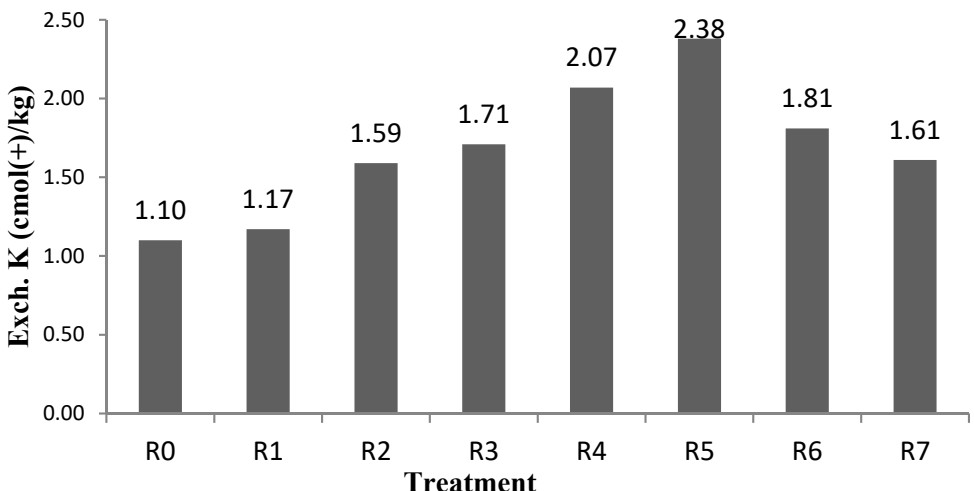

**Figure 5.** Displays the exch. K in the soil after incubation with the combination of red mud and cow manure treatments.

### 3.2.6. Calcium in Soil after Incubation

Table 2 shows that the application of red mud and cow manure fertilizer can increase the exch. Ca in the soil compared to the untreated soil. The exch. Ca content in the untreated soil was 0.24 cmol kg (+) kg$^{-1}$, while the exch. Ca content after treatment ranged from 4.81 cmol kg (+) kg$^{-1}$ to 25.55 cmol kg (+) kg$^{-1}$. The R5 treatment (combination of 70% Ultisol soil + 10% red mud + 20% cow manure fertilizer) showed the highest exch. Ca in the soil. However, it was not significantly different from the R3 treatment (85% Ultisol soil + 5% red mud + 10% cow manure fertilizer) and the R4 treatment (80% Ultisol soil + 10% red mud + 10% cow manure fertilizer). This is because red mud contains 4.22 cmol (+) kg$^{-1}$ of Ca, which is higher than the content of other base cations (K, Mg, and Na), making it a potential source of Ca. Additionally, pH and CEC of the soil are essential factors in determining the exch. Ca in the soil [57,58]. The effect of the combination of red mud and cow manure fertilizer treatments on the exch. Ca in the soil is shown in Figure 6.

### 3.2.7. Magnesium in Soil after Incubation

Table 2 shows that the application of red mud and cow manure fertilizer can increase the exch. Mg in the soil compared to the untreated soil [59,60]. The exch. Mg content in the untreated soil was 0.04 cmol kg (+) kg$^{-1}$, while the exch. Mg content after treatment ranged from 2.65 cmol kg (+) kg$^{-1}$ to 4.07 cmol kg (+) kg$^{-1}$. The R3 treatment (85% Ultisol soil + 5% red mud + 10% cow manure fertilizer) did not differ significantly from the R4 treatment (80% Ultisol soil + 10% red mud + 10% cow manure fertilizer) and the R5 treatment (70% Ultisol soil + 10% red mud + 20% cow manure fertilizer) in terms of exch. Mg in the soil. However, the highest exch. Mg was observed in the R7 treatment (60% Ultisol soil + 15% red mud + 25% cow manure fertilizer), although it did not differ significantly from the R6 treatment (70% Ultisol soil + 15% red mud + 15% cow manure fertilizer). The exch. Mg in the soil increased with the increasing application of red mud and

cow manure fertilizer. This is because red mud contains Mg, which can serve as a source of Mg. The effect of the combination of red mud and cow manure fertilizer treatments on the exch. Mg in the soil is shown in Figure 7.

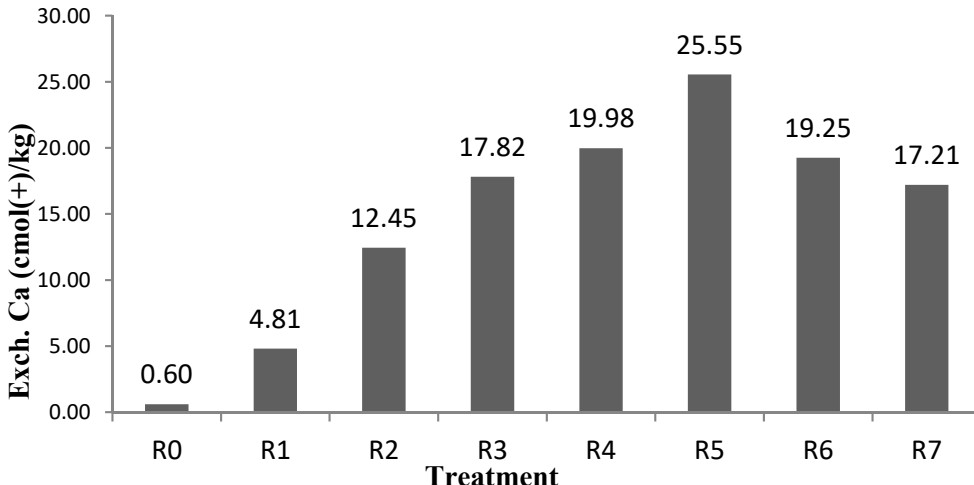

**Figure 6.** Displays the exch. Ca in the soil after incubation with the combination of red mud and cow manure treatments.

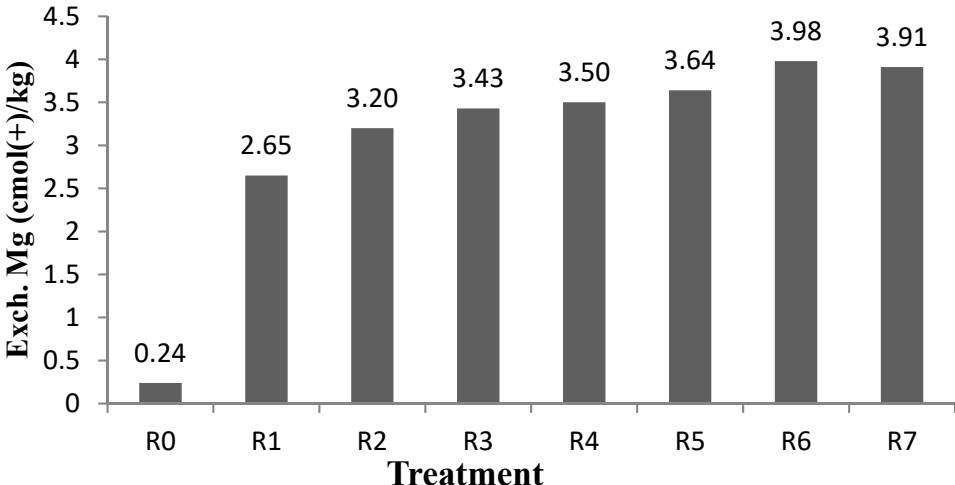

**Figure 7.** Displays the exch. Mg in the soil after incubation with the combination of red mud and cow manure treatments.

### 3.2.8. Sodium in Soil after Incubation

Table 2 shows that the application of red mud and cow manure fertilizer can increase the exch. Na in the soil compared to the untreated soil [61]. The exch. Na content in the untreated soil was 2.25 cmol kg (+) kg$^{-1}$, while the exch. Na content after treatment ranged from 5.20 cmol kg (+) kg$^{-1}$ to 8.41 cmol kg (+) kg$^{-1}$. The R4 treatment (80% Ultisol soil + 10% red mud + 10% cow manure fertilizer) did not differ significantly from the R5 treatment (70% Ultisol soil + 10% red mud + 20% cow manure fertilizer), R6 treatment (70% Ultisol soil + 15% red mud + 15% cow manure fertilizer), and R7 treatment (60% Ultisol soil + 15% red mud + 25% cow manure fertilizer) after incubation. The exch. Na in the soil increased with the increasing application of red mud and cow manure fertilizer up to 15% and 25%. This is because red mud and cow manure fertilizer contain Na, which can serve as a source of exch. Na in the soil. However, Na is weakly retained by organic or mineral matter [62,63]. The effect of the combination of red mud and cow manure fertilizer treatments on the exch. Na in the soil is shown in Figure 8.

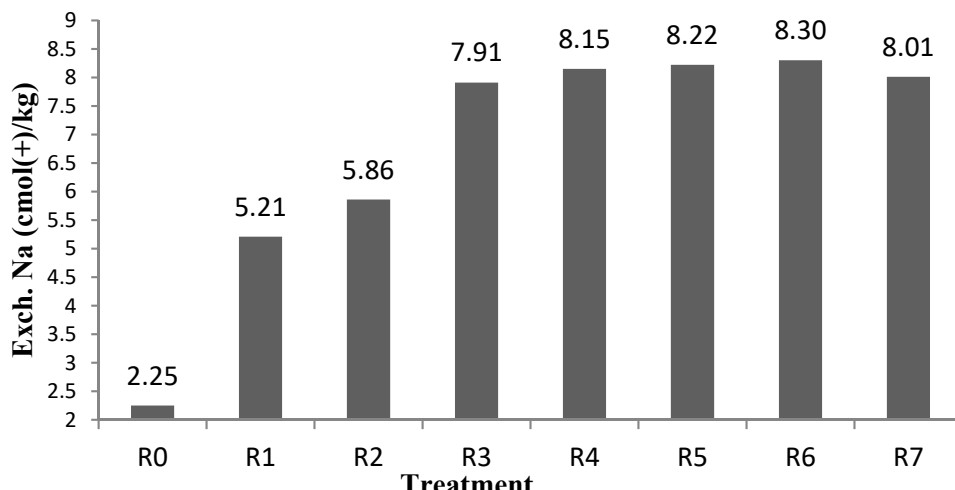

**Figure 8.** Displays the exch. Na in the soil after incubation with the combination of red mud and cow manure fertilizer treatments.

### 3.2.9. Cation Exchange Capacity (CEC) in Soil after Incubation

Table 2 shows that the application of red mud and cow manure fertilizer can increase the cation exchange capacity (CEC) of the soil compared to the untreated soil [64]. The CEC content in the untreated soil was 6.78 cmol kg (+) $kg^{-1}$, while the CEC content after treatment ranged from 10.43 cmol kg (+) $kg^{-1}$ to 13.28 cmol kg (+) $kg^{-1}$. The R2 treatment (95% Ultisol soil + 5% red mud) increased the CEC of the soil after incubation compared to without the addition of red mud. However, the R7 treatment (60% Ultisol soil + 15% red mud + 25% cow manure fertilizer) significantly decreased the CEC of the soil. The highest CEC value was observed in the R5 treatment (10% red mud + 20% cow manure fertilizer), although it did not differ significantly from the R4 treatment (10% red mud + 10% cow manure fertilizer) and R6 treatment (70% Ultisol soil + 15% red mud + 15% cow manure fertilizer) [65]. The increase in CEC of the soil was due to the increasing application of cow manure fertilizer, which has a high organic carbon content (34.22%) that can increase the organic carbon in the soil. The CEC value of the soil is directly related to the amount of organic matter in the soil [65]. One source of negative charge in the soil is organic matter, which comes from the dissociation of functional groups of organic acids. The CEC of humus can reach 150–300 cmol (+) $kg^{-1}$ [66]. The effect of the combination of red mud and cow manure fertilizer treatments on the CEC of the soil is shown in Figure 9.

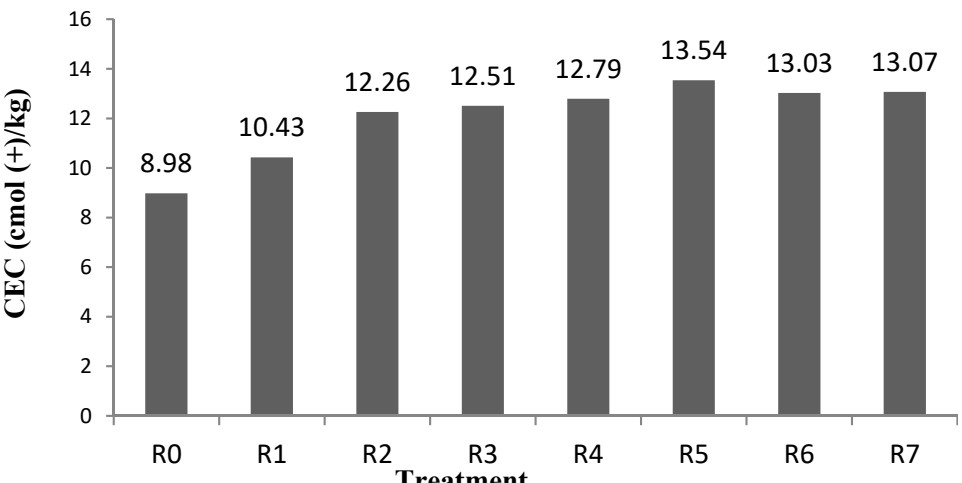

**Figure 9.** Illustrates the cation exchange capacity (CEC) of soil after incubation with a combination of red mud and cow manure.

### 3.2.10. Base Saturated (BS) Soil after Incubation

Table 2 shows that the application of red mud and cow manure fertilizers can significantly increase the soil's base saturated (BS) compared to the untreated soil. The BS value of the untreated soil was 46.65 cmol kg (+) kg$^{-1}$, which increased to a range of 132.67 cmol kg (+) kg$^{-1}$ to 293.83 cmol kg (+) kg$^{-1}$ after treatment. Treatment R2 (95% Ultisol soil + 5% red mud) was found to increase the soil's BS after incubation compared to the untreated soil. However, significant increases in BS were observed with the addition of red mud up to 15% and cow manure fertilizer at 25% kg (treatment R7). This can be attributed to the presence of basic cations such as K, Ca, Mg, and Na in red mud, which contributes to its high BS value of 96.83%, compared to the untreated Ultisol soil BS of approximately 3.7%. Base saturation (BS) is a measure of the percentage of total cation exchange capacity occupied by basic cations such as K, Ca, Mg, and Na [67,68]. The results suggest that red mud and cow manure fertilizers can be used as effective soil amendments to improve soil fertility and productivity by increasing the soil's BS value (Figure 10).

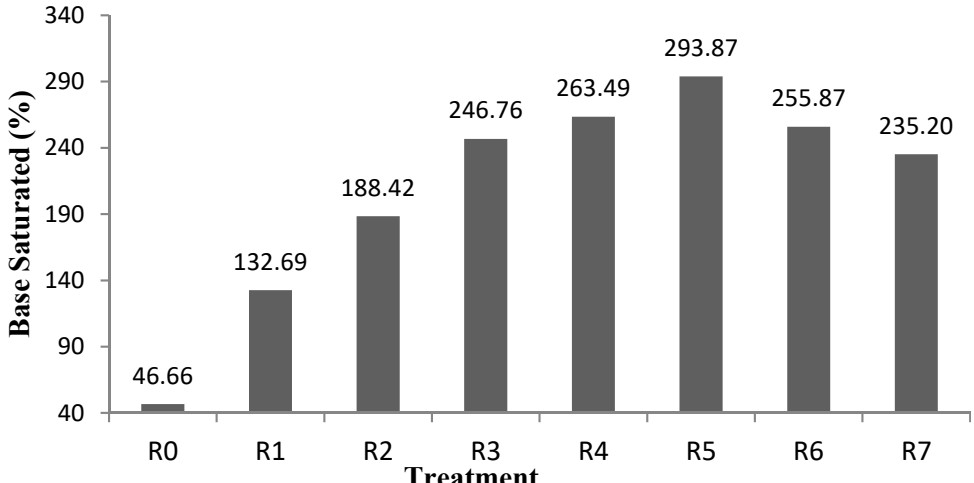

**Figure 10.** Shows the base saturated (BS) of the soil after incubation with a combination of red mud and cow manure fertilizers.

### 3.3. Growth Parameters of Albizia falcata Trees

#### 3.3.1. Plant Height Increment of *Albizia falcata* at Week 10 after Planting

The growth parameters, including plant height and stem diameter (average of six *Albizia falcata* seedlings per treatment), are measurements of the increase in height and stem diameter compared to the initial measurements of the *Albizia falcata* seedlings used (height of 62 cm and diameter of 0.40 cm). This study evaluates the impact of red mud and cow manure fertilizers on the height and stem diameter of *Albizia falcata* trees. The research findings are presented in Table 3.

Table 3 shows that treatment R1 (95% Ultisol soil + 5% red mud) increased the height of *Albizia falcata* trees compared to the control, although there was no significant difference with treatment R2 (90% Ultisol soil + 5% red mud + 5% cow manure fertilizer). Treatment R5 (70% Ultisol soil + 10% red mud + 20% cow manure fertilizer) resulted in the highest height increment, as the N content in the *Albizia falcata* tree tissues was also highest in this treatment. High nitrogen content can improve vegetative growth (height, stem diameter, leaf number, leaf area, shoot number, root number, and root length) [69,70] because nitrogen functions to increase leaf number and area [71]. In addition, the leaf surface area also affects the photosynthesis process [70,72].

The application of red mud and cow manure fertilizers can increase the height of *Albizia falcata* trees up to a certain dosage. There was a tendency for a decrease in tree height with increasing doses of red mud and cow manure fertilizers as the macronutrient levels

that play a role in plant growth also decreased. The effect of combined red mud and cow manure fertilizer treatments on *Albizia falcata* tree height increment can be seen in Figure 11.

**Table 3.** Plant height and stem diameter increment due to the influence of red mud and cow manure fertilizer treatment.

| Treatment | Growth Performance Measurements of *Albizia falcata* Trees | |
|---|---|---|
| | **Plant Height Increment of *Albizia falcata* (cm)** | **Stem Diameter Increment of *Albizia falcata* (cm)** |
| R0 | 2.30 defg | 0.40 gh |
| R1 | 3.17 gh | 0.42 gh |
| R2 | 7.07 efg | 0.57 gh |
| R3 | 11.47 cde | 1.07 ef |
| R4 | 23.40 b | 3.47 a |
| R5 | 32.00 a | 3.63 a |
| R6 | 13.53 c | 2.23 cd |
| R7 | 13.97 c | 2.90 b |

Remarks: The mean followed by the same letter indicates no significant difference in the same row. Source: analysis results.

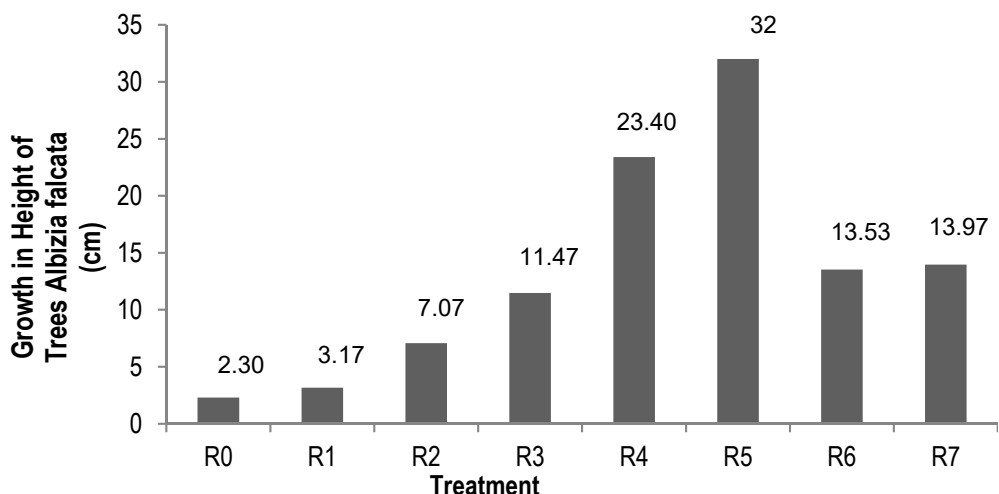

**Figure 11.** Plant height increment of *Albizia falcata* at Week 10 after planting.

3.3.2. Stem Diameter Increment of *Albizia falcata* at Week 10 after Planting

Table 2 shows that treatment R1 (95% Ultisol soil + 5% Red Mud) increased the height of *Albizia falcata* trees compared to the control, although there was no significant difference with treatments R2 (90% Ultisol soil + 5% red mud + 5% cow manure fertilizer). The present study investigated the effects of red mud and cow manure on the growth of plants, specifically in terms of stem diameter. Results showed that the R5 treatment, consisting of Ultisol soil (70%), red mud (10%), and cow manure fertilizer (20%), resulted in the highest and significantly different increase in stem diameter compared to other treatments, except for the R4 treatment (red mud 10% + cow manure 10%). The combination of red mud and cow manure in the R5 treatment was found to promote stem diameter growth. However, there was a decreasing trend in stem diameter growth with increasing amounts of red mud and cow manure until the R7 treatment (red mud 15% + cow manure 25%). This could be attributed to a reduction in macronutrient levels, which play a crucial role in plant growth. Overall, the results suggest that the R5 treatment is the most effective in promoting stem diameter growth in plants.

Sufficient available nutrients in the soil can affect plant physiology and metabolism processes better and improve plant growth [73,74]. Moreover, an increase in soluble nutrients in the soil can increase the absorption of nutrients by plants for their growth [75,76]. The effect of combined red mud and cow manure fertilizer treatments on *Albizia falcata* tree stem diameter increment can be seen in Figure 12.

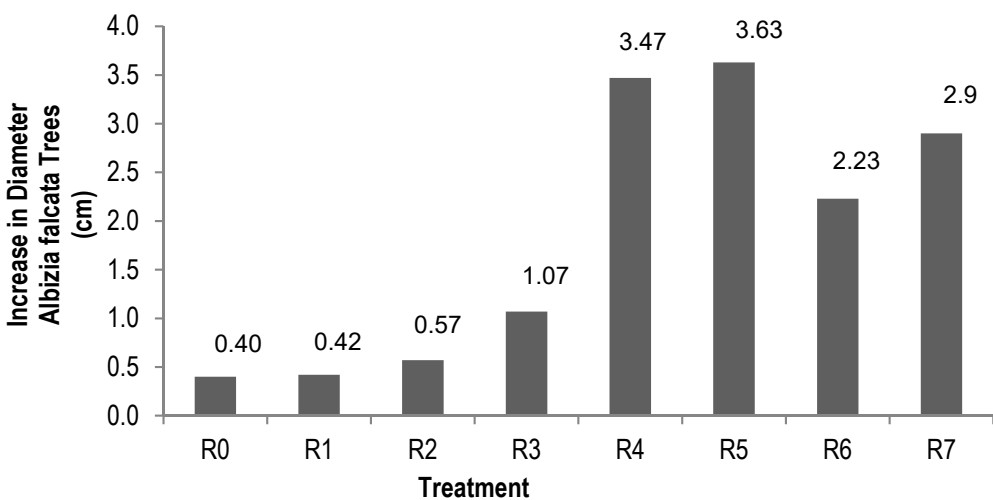

**Figure 12.** Stem diameter increment of *Albizia falcata* at Week 10 After planting.

### 4. Conclusions

Sustainable rehabilitation of post-bauxite mining land can be successfully executed by employing site-specific amelioration technologies, which capitalize on red mud waste and organic compounds such as cow manure fertilizer. This strategy serves as an efficient tool for mitigating the adverse effects associated with bauxite mining and processing. By mitigating land degradation and curbing the accumulation of red mud waste, this approach simultaneously boosts nutrient availability and encourages the proliferation of *Albizia falcata* plants.

The findings of this study show that an amalgamation of 10% red mud and 20% cow manure fertilizer significantly amplified soil pH levels by over twice the initial value (an increase of 97.65%). This intervention also resulted in improved availability of key nutrients (N, P, K, Ca, Mg, and Na), with an observed increase ranging from 2 to 60 times (an increment of 116.36% to 5966.67%). This led to a substantial improvement in the growth of *Albizia falcata* plants by factors ranging from 8 to 13 times (an increase of 807.50% to 1291.30%) when compared to the control in post-bauxite mining land.

The focus of this study was exclusively on the rehabilitation of post-bauxite mining land in PT Antam, Sanggau Regency, West Kalimantan. The study involved *Albizia falcata* plants and used red mud and cow manure as the chosen ameliorants. Parameters observed were confined to macro-nutrient availability and plant growth factors, including plant height and stem diameter. The duration of the research was one year. Nevertheless, it is recommended that future research expands the area of study, includes additional plant species, examines treatments with different ameliorants, and broadens the scope of observed variables. For a holistic understanding and to evaluate the sustainability and economic potential of this rehabilitation method, a more exhaustive environmental assessment, coupled with a cost-benefit analysis with respect to the government, the industry, and the local community, is advised.

**Author Contributions:** Writing—original draft, D.S.; Writing—review & editing, N.I.D. All authors have read and agreed to the published version of the manuscript.

**Funding:** This research was funded by the Directorate General of Higher Education (DIKTI) grant number: 101/SP2H/LT/DRPM/2019.

**Data Availability Statement:** Data will be made available on request.

**Acknowledgments:** The authors would like to express their gratitude to the Directorate General of Higher Education (DIKTI) for funding this research through the National Priority Research Program of MP3EI (Masterplan for the Acceleration and Expansion of Indonesia's Economic Development), which enabled this research to be conducted.

**Conflicts of Interest:** The authors declare no conflict of interest.

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
