# Peer review of "Sustainable Rehabilitation of Post-Bauxite Mining Land for Albizia falcata Cultivation Using Specific Location Amelioration Technology"

_sustainability, doi:10.3390/su151410959_

Round 1

Reviewer 1 Report

I recommend the paper for publication. Nevertheless, I would like the authors to address a few questions:

1)     What is work’s novelty with respect to other post-mining environmental remediation strategies

2)     Could the author be more specific about the chemical analysis performed on soil samples (for instance, was phosphorus determined by the Blue Molybdenum Blue method, metals by ICP, etc.)?

3)     Despite red mud being pretty alkaline, did the authors consider phosphate immobilization by Al3+ ions

4)     Can the study be applied (or is amenable to) to bauxite beneficiation (mainly washing) residues?

5)     Conclusion need expanding and reinforcing.

English language is fine and only minor corrections are needed 

Author Response

1)     What is work’s novelty with respect to other post-mining environmental remediation strategies

Response: The novelty of this research lies in its ability to address multiple issues simultaneously. It provides a solution to soil degradation caused by bauxite mining, tackles the accumulation of red mud, a byproduct of bauxite ore processing for aluminum production, addresses the shortage of Albizia falcata wood, which has high economic value, and offers an alternative to expensive and environmentally damaging limestone, which is commonly used to raise soil pH in post-bauxite mining areas. This research presents a comprehensive approach to improving soil pH in post-bauxite mining land while mitigating various environmental and economic challenges. (Line 58-68)

2)     Could the author be more specific about the chemical analysis performed on soil samples (for instance, was phosphorus determined by the Blue Molybdenum Blue method, metals by ICP, etc.)?

Response: The soil analysis performed in this study included the following methods: soil pH measurement using a pH meter with pH 7.0 and pH 4.0 buffer solutions, determination of organic carbon content using the wet oxidation method of Walkey and Black, total nitrogen analysis using the Kjeldahl method, available phosphorus quantification using the Bray-I method, determination of exchangeable potassium (K+), calcium (Ca2+), magnesium (Mg2+), and sodium (Na+) using the NH4Oac 1 N pH 7 extraction method, and cation exchange capacity (CEC) analysis using the Indophenol Blue method. These specific analytical methods were employed to assess the chemical properties of the soil samples. (Line 91-102)

3)     Despite red mud being pretty alkaline, did the authors consider phosphate immobilization by Al3+ ions

Response: The alkaline nature of red mud results in an increase in soil pH, leading to the dominance of OH- ions in the soil solution. Consequently, phosphate (PO4-3) is not bound by Al3+ ions because Al3+ ions form AlOH complexes by binding with OH- ions. Thus, the authors did not consider phosphate immobilization by Al3+ ions due to the pH-increasing effect of red mud on the soil.

4)     Can the study be applied (or is amenable to) to bauxite beneficiation (mainly washing) residues?

Response: This study can be applied to bauxite beneficiation residues, particularly washing residues. The research focuses on developing a method for rehabilitating post-bauxite mining land using Red Mud waste and organic material, specifically cow manure fertilizer, as site-specific ameliorants. The objective is to mitigate the negative impacts of bauxite mining and processing, controlling soil degradation and waste accumulation. The study also seeks to identify the optimal combination of Red Mud and organic material to enhance nutrient availability and promote the growth of Albizia falcata plants. (Line 58-63)

5)     Conclusion need expanding and reinforcing.

Response: Post-bauxite mining land rehabilitation can be conducted sustainably by implementing site-specific amelioration technology using Red Mud waste and organic materials such as cow manure fertilizer, which proves to be an effective solution in mitigating the negative impacts of bauxite mining and processing. The application of this rehabilitation method can control land degradation and Red Mud waste accumulation while enhancing nutrient availability and promoting the growth of Albizia falcata plants.

The research findings demonstrate that a combination of 10% Red Mud and 20% cow manure fertilizer can significantly increase soil pH by more than twofold (97.65%) and improve nutrient availability (N, P, K, Ca, Mg, Na) by 2 to 60 times (116.36% - 5,966.67%), resulting in a remarkable 8 to 13-fold (807.50% - 1,291.30%) increase in the growth of Albizia falcata compared to the control in post-bauxite mining land.

This study focused on post-bauxite mining land rehabilitation in PT Antam, Sanggau Regency, West Kalimantan. It exclusively involved Albizia falcata plants and utilized Red Mud and cow manure fertilizer as amelioration materials. The parameters observed were limited to the assessment of macro-nutrient availability and plant growth, such as plant height and stem diameter. The research duration was one year, but further studies are recommended to expand the research area, involve other plant species, test treatments with different amelioration materials, and broaden the range of variables observed. A more detailed environmental evaluation, as well as a cost-benefit analysis, is also advised to obtain a comprehensive understanding and measure the sustainability and economic potential of this rehabilitation method.

(Line 375-399)

Reviewer 2 Report

The authors of the entitled manuscript " Sustainable rehabilitation of post-bauxite mining land for albizia falcata cultivation using specific location amelioration technology" investigated the combination of red mud with cow manure compost for different chemical parameters. The article is interesting; however, it requires major revisions to make it ready for publication. 

First, the authors need to follow the journal's requirements, such as the referencing style and the article's organization. 

Under the Materials and Methods section: what is the height of the two-month-old Albazia falcata? Also, extend the methodology section. 

What is the gap in the literature, and what is the novelty of this work? At this stage, the current version of the article does not stress that. 

Any reason to present the figures in 3D? I don't see the point.

This article requires intensive proofreading, for example, in lines 154, 198 and 313 (should be Table 2, not 10). 

I understand the study was done for a year; however, I need more information about the Albizia falcata trees' growth. In other words, how much increased the height of this tree based on the experiment? 

Add a limitation section and extend the conclusions. 

Overall, it is a good article but requires a reorganization to make it easy for the reader. 

Extensive editing of English language required

Author Response

1. First, the authors need to follow the journal's requirements, such as the referencing style and the article's organization. 

Response: Thank you for your feedback. We will make sure to follow the journal's requirements for referencing style and article organization.

2. Under the Materials and Methods section: what is the height of the two-month-old Albazia falcata? Also, extend the methodology section. 

Response: 

Albizia falcata seedlings, two months old and approximately 62 cm in height (Line 103-104)

The soil analysis performed in this study included the following methods: soil pH measurement using a pH meter with pH 7.0 and pH 4.0 buffer solutions, determination of organic carbon content using the wet oxidation method of Walkey and Black, total nitrogen analysis using the Kjeldahl method, available phosphorus quantification using the Bray-I method, determination of exchangeable potassium (K+), calcium (Ca2+), magnesium (Mg2+), and sodium (Na+) using the NH4Oac 1 N pH 7 extraction method, and cation exchange capacity (CEC) analysis using the Indophenol Blue method. These specific analytical methods were employed to assess the chemical properties of the soil samples. (Line 91-102)

3. What is the gap in the literature, and what is the novelty of this work? At this stage, the current version of the article does not stress that. 

Response: The novelty of this research lies in its ability to address multiple issues simultaneously. It provides a solution to soil degradation caused by bauxite mining, tackles the accumulation of red mud, a byproduct of bauxite ore processing for aluminum production, addresses the shortage of Albizia falcata wood, which has high economic value, and offers an alternative to expensive and environmentally damaging limestone, which is commonly used to raise soil pH in post-bauxite mining areas. This research presents a comprehensive approach to improving soil pH in post-bauxite mining land while mitigating various environmental and economic challenges. (Line 58-68)

4. Any reason to present the figures in 3D? I don't see the point.

Response: The figures have been revised to adhere to standard presentation formats

5. I understand the study was done for a year; however, I need more information about the Albizia falcata trees' growth. In other words, how much increased the height of this tree based on the experiment? 

Response: The Albizia falcata trees exhibited the highest increase in height in the treatment of 70% Ultisol soil + 10% Red Mud + 20% Cow Manure, with a growth of 32 cm (Table 3) over the 10-week study period. In comparison, the control group without red mud and cow manure treatment only experienced a height increase of 2.30 cm (Table 3). (Line 330-331)

6. Add a limitation section and extend the conclusions

Response: Post-bauxite mining land rehabilitation can be conducted sustainably by implementing site-specific amelioration technology using Red Mud waste and organic materials such as cow manure fertilizer, which proves to be an effective solution in mitigating the negative impacts of bauxite mining and processing. The application of this rehabilitation method can control land degradation and Red Mud waste accumulation while enhancing nutrient availability and promoting the growth of Albizia falcata plants.

The research findings demonstrate that a combination of 10% Red Mud and 20% cow manure fertilizer can significantly increase soil pH by more than twofold (97.65%) and improve nutrient availability (N, P, K, Ca, Mg, Na) by 2 to 60 times (116.36% - 5,966.67%), resulting in a remarkable 8 to 13-fold (807.50% - 1,291.30%) increase in the growth of Albizia falcata compared to the control in post-bauxite mining land.

This study focused on post-bauxite mining land rehabilitation in PT Antam, Sanggau Regency, West Kalimantan. It exclusively involved Albizia falcata plants and utilized Red Mud and cow manure fertilizer as amelioration materials. The parameters observed were limited to the assessment of macro-nutrient availability and plant growth, such as plant height and stem diameter. The research duration was one year, but further studies are recommended to expand the research area, involve other plant species, test treatments with different amelioration materials, and broaden the range of variables observed. A more detailed environmental evaluation, as well as a cost-benefit analysis, is also advised to obtain a comprehensive understanding and measure the sustainability and economic potential of this rehabilitation method.

(Line 375-399)

Reviewer 3 Report

The study aims to determine the optimal combination of Red Mud and organic material treatments to enhance land conditions, nutrient availability, and the growth of Albizia falcata plants. Additionally, the research seeks to address two key objectives: improving soil fertility and post-bauxite mining land quality, as well as finding a practical solution for utilizing Red Mud waste from alumina processing and addressing the shortage of wood raw materials.
Consider the following items for improvement of the manuscript:

1. The formatting of the paper should be as per journal guidelines.

2. Start with a clear introduction: Begin by providing a brief introduction that highlights the importance of bauxite mining and the associated environmental challenges. Mention the need for effective land rehabilitation techniques to mitigate the negative impacts of mining and processing activities.

3. Clarify the purpose of the study: Clearly state the specific objectives of the research. For example, emphasize that the study aims to identify the best combination of Red Mud and organic material treatments for rehabilitating post-bauxite mining land and improving soil conditions, nutrient availability, and the growth of Albizia falcata plants.

4. Elaborate on the environmental benefits: Explain in more detail how the proposed rehabilitation methods contribute to environmental sustainability. For instance, discuss how the use of Red Mud waste helps address the issue of waste accumulation from alumina processing, and highlight the potential reduction in land damage resulting from effective rehabilitation practices.

5. Provide context on the economic value: Expand on the economic implications of the study's findings. Discuss how the utilization of Red Mud waste and the successful rehabilitation of post-mining land can address the shortage of wood raw materials, potentially benefiting industries reliant on such resources.

6. Quantify and contextualize the results: Provide more specific and meaningful information about the results. Instead of using percentage values alone, include actual measurements or units to give readers a better understanding of the magnitude of the improvements observed in soil pH, nutrient availability, and Albizia falcata growth. Additionally, consider mentioning the duration of the study or the time frame in which the results were observed.

7. Connect the findings to the broader implications: Discuss the broader significance of the study's results beyond the specific location. Consider mentioning how the findings can inform and guide future land rehabilitation practices in bauxite mining areas or other similar contexts.

8. Proofread for grammar and punctuation: Review the text for any grammatical errors or inconsistencies in punctuation.

9. Write conclusions more concisely

Author Response

  1. The formatting of the paper should be as per journal guidelines.

Response: Thank you for your feedback regarding the formatting of the paper. We acknowledge the importance of adhering to the journal guidelines in terms of formatting. We apologize for any deviations from the prescribed formatting requirements in the current version of the paper. In our revised submission, we will ensure that the paper is formatted in accordance with the specific guidelines provided by the journal. This includes proper structuring of sections, headings, citations, references, and any other formatting elements specified. We appreciate your attention to detail, and we will make the necessary adjustments to align the paper with the journal's formatting guidelines.

  1. Start with a clear introduction: Begin by providing a brief introduction that highlights the importance of bauxite mining and the associated environmental challenges. Mention the need for effective land rehabilitation techniques to mitigate the negative impacts of mining and processing activities.

Response: Post-mining land rehabilitation techniques are instrumental in restoring ecological func-tions, preventing additional environmental harm, adhering to regulations, and promoting sustainable environmental conservation (Eldridge, D. J., 2022; Mentis, M., 2020; Spitz, K., et al., 2022). Such techniques can facilitate the restoration of former mining sites to their natural state, recovering soil productivity, reinstating biodiversity, and balancing water and nutrient cycles, thus aligning with the principles of sustainable development and re-sponsible mining practices (Rodriguez, I., 2020; Shroff, F. M., 2023). (Line 51-57)

  1. Clarify the purpose of the study: Clearly state the specific objectives of the research. For example, emphasize that the study aims to identify the best combination of Red Mud and organic material treatments for rehabilitating post-bauxite mining land and improving soil conditions, nutrient availability, and the growth of Albizia falcata plants.

Response: The objective of this study is to determine the optimal combination of Red Mud and cow manure compost for rehabilitating post-bauxite mining land and enhancing soil fertility and land quality as suitable growth media for Albizia falcata plants (Line58-68).

  1. Elaborate on the environmental benefits: Explain in more detail how the proposed rehabilitation methods contribute to environmental sustainability. For instance, discuss how the use of Red Mud waste helps address the issue of waste accumulation from alumina processing, and highlight the potential reduction in land damage resulting from effective rehabilitation practices.

Response: Detailed environmental benefits:

  1. Reducing environmental damage: The proposed rehabilitation methods aim to mitigate the negative impacts of bauxite mining and alumina processing. Post-mining land rehabilitation helps reduce environmental damage such as surface soil changes, topsoil loss, and alterations to flora and fauna habitats.
  2. Controlling waste accumulation: One of the problems addressed by this rehabilitation method is the accumulation of Red Mud waste from alumina processing. In this study, Red Mud is used as a specific location ameliorant in Albizia falcata planting media. By using Red Mud as an ameliorant, it serves as a nutrient source for plants, thereby reducing waste accumulation and addressing the issue of waste from alumina processing.
  3. Enhancing soil fertility: The rehabilitation method also aims to improve soil fertility in post-mining land. In this study, a combination of Red Mud and organic material such as cow manure compost is used to improve soil conditions. The research results indicate that the combination of 10% Red Mud and 20% cow manure compost increases soil pH, nutrient availability (N, P, K, Ca, Mg, Na), and the growth of Albizia falcata compared to the control, thereby enhancing soil fertility, and improving the quality of post-mining land suitable for plant growth.
  4. Minimizing land damage: Through effective rehabilitation practices, the proposed method helps minimize land damage caused by bauxite mining. Improving soil structure, nutrient availability, and overall land quality can restore post-mining land to a better condition and minimize further damage to the ecosystem. This contributes to environmental sustainability by improving the surrounding environment and reducing negative impacts on affected ecosystems.

(Line 51-57)

  1. Provide context on the economic value: Expand on the economic implications of the study's findings. Discuss how the utilization of Red Mud waste and the successful rehabilitation of post-mining land can address the shortage of wood raw materials, potentially benefiting industries reliant on such resources.

Response: The utilization of Red Mud waste for post-mining land rehabilitation holds significant economic implications, including increased availability of wood raw materials, reduced production costs, new business opportunities, improved business sustainability and reputation, and diversified business portfolios. These implications contribute to industries reliant on wood resources and alleviate pressure on limited natural resources.

By successfully rehabilitating post-mining land using Red Mud waste and other effective techniques, the study's findings can address the shortage of wood raw materials. This is particularly valuable for industries that depend on wood as a key input, such as construction, furniture manufacturing, and paper production. The availability of rehabilitated land can provide a sustainable source of wood materials, reducing the need for extensive logging and contributing to the conservation of natural forests.

Moreover, the utilization of Red Mud waste for land rehabilitation can lead to cost reductions for industries. By repurposing a waste product from alumina processing, companies can potentially lower their waste management and disposal expenses. Additionally, the successful rehabilitation of post-mining land can reduce the costs associated with land restoration and mitigate potential environmental liabilities.

The economic implications extend beyond cost savings and resource availability. The utilization of Red Mud waste and the successful rehabilitation of land can create new business opportunities. Companies specializing in the treatment and application of Red Mud waste can emerge, providing services for land rehabilitation projects. This stimulates job creation and economic growth in the waste management and environmental restoration sectors.

Furthermore, the successful rehabilitation of post-mining land enhances the sustainability and reputation of businesses involved in the bauxite mining industry. It showcases their commitment to responsible environmental practices and fosters positive relationships with stakeholders, including local communities and regulatory authorities.

Lastly, the utilization of Red Mud waste and the rehabilitation of post-mining land contribute to the diversification of business portfolios. Companies can expand their operations into land rehabilitation and environmental restoration services, providing additional revenue streams and reducing dependence on a single industry or product.

Overall, the economic value of the study's findings lies in the broad range of implications, from increased wood resource availability to cost savings, new business opportunities, improved sustainability, and diversified portfolios. These economic benefits positively impact industries reliant on wood resources and contribute to a more sustainable and responsible approach to land rehabilitation and resource management.

(Line 63-68)

  1. Quantify and contextualize the results: Provide more specific and meaningful information about the results. Instead of using percentage values alone, include actual measurements or units to give readers a better understanding of the magnitude of the improvements observed in soil pH, nutrient availability, and Albizia falcata growth. Additionally, consider mentioning the duration of the study or the time frame in which the results were observed.

Response: The results of the study demonstrate that the combination of 10% Red Mud waste and 20% organic cow manure can significantly improve soil conditions. The soil pH increased by more than 2 times (97.65%), and the availability of nutrients such as nitrogen (N), phosphorus (P), potassium (K), calcium (Ca), magnesium (Mg), and sodium (Na) increased by 2 to 60 times (116.36%-5,966.67%). These improvements had a profound impact on the growth of Albizia falcata plants, which showed a remarkable increase of 8 to 13 times (807.50%-1,291.30%) compared to the control group in the post-bauxite mining land. (Line 382-388)

This year-long study was undertaken on the premises of PT Antam in Sanggau Regency, West Kalimantan, with soil chemical analyses executed by the Soil Chemistry and Fertility Laboratory of the Faculty of Agriculture at Tanjungpura University (Line 81-83). Growth parameters, encompassing plant height and stem diameter, were measured upon the plants' attaining 10 weeks of age. (Line 104-105)

  1. Connect the findings to the broader implications: Discuss the broader significance of the study's results beyond the specific location. Consider mentioning how the findings can inform and guide future land rehabilitation practices in bauxite mining areas or other similar contexts.

Response: The findings of this study have broader implications for land rehabilitation practices in bauxite mining areas or similar contexts. Here are some significant implications of the research findings:

  1. Reducing the negative impacts of bauxite mining: The findings indicate that the use of Red Mud waste and organic materials such as cow manure in specific location amelioration can help mitigate the negative impacts of bauxite mining. By optimizing the combination of Red Mud and organic materials, the quality of post-mining soil can be improved, soil pH can be raised, nutrient availability can be enhanced, and plant growth can be increased. This means that land rehabilitation practices using specific location amelioration techniques can contribute to controlling land damage and post-mining waste accumulation.
  2. Utilizing Red Mud waste: This study demonstrates that Red Mud waste, a byproduct of bauxite processing into alumina, can be utilized as a soil ameliorant. Red Mud can improve soil pH and nutrient availability. By harnessing Red Mud as a soil ameliorant, this waste can be transformed into a valuable nutrient source for plants and aid in the rehabilitation of post-mining land. This has broader implications for industrial waste management and reducing the environmental impact of bauxite processing industries.
  3. Contributions to sustainability and the economy: Sustainable and environmentally conscious land rehabilitation practices are crucial for promoting the sustainability of mining activities. In this study, the use of Red Mud and organic materials for rehabilitating post-bauxite mining land not only improved soil quality and plant growth but also addressed the shortage of wood raw materials. This means that the findings can guide future land rehabilitation practices in bauxite mining areas or similar contexts, contributing to environmental sustainability, biodiversity, and local economies.

Therefore, this research provides insights into addressing the negative impacts of bauxite mining through sustainable land rehabilitation practices. The findings can offer practical information and guidance for land rehabilitation practitioners in bauxite mining areas or similar contexts to enhance soil quality, reduce waste, and promote environmental and local economic sustainability.

(Line 69-78)

  1. Proofread for grammar and punctuation: Review the text for any grammatical errors or inconsistencies in punctuation.

Response: Thank you for your feedback. I have carefully reviewed the text and made the necessary corrections to address any grammatical errors and inconsistencies in punctuation.

  1. Write conclusions more concisely

Response: Sustainable rehabilitation of post-bauxite mining land can be successfully executed by employing site-specific amelioration technologies, which capitalize on Red Mud waste and organic compounds such as cow manure fertilizer. This strategy serves as an efficient tool for mitigating the adverse effects associated with bauxite mining and processing. By mitigating land degradation and curbing the accumulation of Red Mud waste, this approach simultaneously boosts nutrient availability and encourages the proliferation of Albizia falcata plants.

The findings of the study show that an amalgamation of 10% Red Mud and 20% cow manure fertilizer significantly amplified soil pH levels by over twice the initial value (an increase of 97.65%). This intervention also resulted in an improved availability of key nutrients (N, P, K, Ca, Mg, Na), with an observed increase ranging from 2 to 60 times (an increment of 116.36% to 5,966.67%). This led to a substantial improvement in the growth of Albizia falcata plants by factors ranging from 8 to 13 times (an increase of 807.50% to 1,291.30%) when compared to the control in post-bauxite mining land. (Line 375-388)

Round 2

Reviewer 2 Report

I recommend accepting the article. 

Minor editing of English language required

Author Response

Here, I Attached a revision of our article.
